# Mitigation of Acetamiprid Residue Disruption on Pea Seed Germination by Selenium Nanoparticles and Lentinans

**DOI:** 10.3390/plants12152781

**Published:** 2023-07-27

**Authors:** Yongxi Lin, Chunran Zhou, Dong Li, Yujiao Jia, Qinyong Dong, Huan Yu, Tong Wu, Canping Pan

**Affiliations:** 1Innovation Center of Pesticide Research, Department of Applied Chemistry, College of Science, China Agricultural University, Beijing 100193, China; lyx945748557@126.com (Y.L.);; 2Huizhou Yinnong Technology Co., Ltd., Huizhou 516057, China; 3Key Laboratory of Green Prevention and Control of Tropical Plant Diseases and Pests, College of Plant Protection, Ministry of Education, Hainan University, Haikou 570228, China

**Keywords:** germination, pesticide stress, exogenous substances, phenylpropanoid pathway, quality

## Abstract

The use of pesticides for pest control during the storage period of legume seeds is a common practice. This study evaluated the disruptive effects on pea seed germination and the repair effects of selenium nanoparticles (SeNPs) and lentinans (LNTs) This study examined the biomass, nutrient content, antioxidant indicators, plant hormones, phenolic compounds, and metabolites associated with the lignin biosynthesis pathway in pea sprouts. The application of acetamiprid resulted in a significant decrease in yield, amino-acid content, and phenolic compound content of pea sprouts, along with observed lignin deposition. Moreover, acetamiprid residue exerted a notable level of stress on pea sprouts, as evidenced by changes in antioxidant indicators and plant hormones. During pea seed germination, separate applications of 5 mg/L SeNPs or 20 mg/L LNTs partially alleviated the negative effects induced by acetamiprid. When used in combination, these treatments restored most of the aforementioned indicators to levels comparable to the control group. Correlation analysis suggested that the regulation of lignin content in pea sprouts may involve lignin monomer levels, reactive oxygen species (ROS) metabolism, and plant hormone signaling mediation. This study provides insight into the adverse impact of acetamiprid residues on pea sprout quality and highlights the reparative mechanism of SeNPs and LNTs, offering a quality assurance method for microgreens, particularly pea sprouts. Future studies can validate the findings of this study from the perspective of gene expression.

## 1. Introduction

Pea sprouts are tender leaves and stems that emerge from germinated pea seeds [1]. Pea sprout as a microgreen is popular with consumers for its delicate texture, pleasant fragrance, and emerald green color [2]. Pea sprouts are not only tasty and visually appealing, but also packed with valuable nutrients that can provide a range of health benefits [3]. Pea sprouts are rich in soluble sugars, soluble proteins, amino acids, and phenols [4,5,6]. Phenols, in particular, can help protect against cellular damage and reduce the risk of chronic diseases in human bodies [7].

Pea seeds are vulnerable to damage from pests, such as *Bruchus pisorum*, during the storage period, and pesticides may be utilized to prevent pest damage [8,9]. The use of pesticides can result in residues that can pose a dietary risk for consumers [10]. In addition, pesticide use can affect the growth and nutritional quality of the pea sprouts that grow from the seeds [11]. Pea sprouts are susceptible to various environmental factors, including temperature, light, humidity, plant pathogens, and environmental contaminants [12], which can lead to lignification [13], a condition where the sprouts become tough and fibrous, making them less palatable and nutritious [14,15].

Microgreens, such as pea sprouts, are an excellent source of green vegetables due to their short production cycle and minimal pesticide usage [3]. Although microgreens may not be directly exposed to pesticides during production, the seeds from which they sprout may have been treated with pesticides during storage. The use of pesticides on pea seeds may lead to dietary risks and can impact the quality of the resulting pea sprouts. Previous research [16,17] on microgreens has primarily focused on the impact of microbiological contaminants and toxins on nutritional qualities of microgreens. Wongmaneepratip and Yang [18] investigated the migration patterns of pyrethroid insecticides between the growth medium (water and soil) and mung bean sprouts. Although pesticide usage during seed storage is a common practice in many regions [19], few studies have explored studies on the potential impact of pesticide residues on the quality of sprouts grown from treated seeds. Nanoparticles [20,21] and polysaccharides [22] are frequently employed to activate plant defense systems in response to various stressors. However, there is limited research on the combined application of these two agents and their effects.

The purpose of this study was to investigate the impact of acetamiprid (ACE), a widely used insecticide known for its fast-acting and low-residue properties, on the quality of pea sprouts produced from stored seeds treated with this insecticide. Additionally, the study also aimed to evaluate the potential of selenium nanoparticles (SeNPs), lentinans (LNTs), and their combination to remediate the quality loss induced by acetamiprid. The residue behavior of acetamiprid during pea seed storage and sprout production was analyzed. Parameters including morphological characteristics, nutritional contents, antioxidant capacity, plant hormones, polyphenols, and the lignin pathway were assessed to evaluate the quality of pea sprouts. Overall, the study provides a feasible solution for quality management and assurance in the production process of pea sprouts.

## 2. Results

### 2.1. Degradation and Residue of Acetamiprid on Pea Seeds

The trend of acetamiprid residual dissipation in pea seeds is depicted in Figure 1. Overall, the dissipation dynamic of acetamiprid in pea seeds was fitted with a first-order kinetic equation (C = C_0_e^−kt^). The half-life of acetamiprid degradation is 4.18 days (ln2/k). At 2 h after acetamiprid spraying, the deposition of acetamiprid on pea seeds was 1.58 mg/kg. By the 14th day after acetamiprid spraying, the concentration of acetamiprid on pea seeds decreased to 0.18 mg/kg, and the degradation rate reached 88.6%.

After washing, disinfection, and soaking, the residual concentration of acetamiprid in pea seeds was determined to be below the LOQ (<0.01 mg/kg) of the instrument, which inferred a low probability of risk associated with the presence of acetamiprid residues during the cultivation and processing of pea sprouts.

### 2.2. Growth and Morphometric Parameters

The effects of acetamiprid residues on pea germination and growth were investigated, and the results are presented in Figure 2. Despite low levels of acetamiprid residues on the pea seeds after disinfection, cleaning, and soaking, acetamiprid had a negative impact on the morphologies of the pea sprouts. As shown in Figure 2A, the hypocotyls of the pea sprouts were significantly inhibited when treated with acetamiprid on the first day of germination. Furthermore, as evidenced by Figure 2B and Table 1, the plant height and edible fresh weight of the pea sprouts treated with acetamiprid were substantially lower than those of the control group on the eighth day of germination. Following exogenous spraying of SeNPs and LNTs, the plant height and edible fresh weight recovered to some extent. Notably, when used in combination, the plant height and edible fresh weight of the pea sprouts recovered to levels comparable with those of the control group.

### 2.3. Nutrient Contents

The application of acetamiprid did not have a significant impact on the soluble sugar (Figure 3A) and soluble protein content (Figure 3B), but it did result in a significant reduction of 54.0% in the amino-acid content (Figure 3C). However, after the application of SeNPs and LNTs, there was a partial restoration of amino-acid content, with increases of 78.4% and 39.2%, respectively. Furthermore, the amino-acid content returned to the control level with no significant difference after the combined treatment with SeNPs and LNTs (Figure 3C). Additionally, the combined application of SeNPs and LNTs led to a significant increase in the soluble sugar content (Figure 3A). These findings suggest that the combined application of SeNPs and LNTs may be an effective approach to mitigate the negative effects of acetamiprid on amino acid.

### 2.4. Antioxidant Indexes

From Figure 4, it can be seen that application of acetamiprid on pea seeds during storage affected antioxidant defense systems in pea sprouts after germination.

After treatment with acetamiprid, the activities of SOD and CAT, as well as the MDA content of pea sprouts, were significantly increased compared to the control group, with respective increases of 28.5%, 15.8%, and 23.9% (Figure 4A,C,D). However, the activity of POD, the level of GSH, and the DPPH^●^-scavenging rate were significantly reduced, with respective decreases of 28.4%, 19.6%, and 36.8% (Figure 4B,E,F). When SeNPs and LNTs were applied in combination, the SOD activity and GSH content were restored to levels similar to those of the control group, but there was no significant difference in the SOD activity and GSH content compared with the acetamiprid-treated group when applied alone (Figure 4A,D). LNTs applied alone significantly increased the POD activity, but it was still lower than that of the control group. After combined application with SeNPs, the POD was increased to a level similar to that of the control group (Figure 4B). CAT activity could be reduced to a level similar to that of the control group by applying SeNPs, LNTs, or both (Figure 4C). SeNPs and LNTs applied alone could significantly reduce and increase the MDA content and DPPH^●^-scavenging rate, respectively, but there were still significant differences compared with the control group. When applied in combination, the MDA content and DPPH^●^-scavenging rate were restored to levels similar to those of the control group (Figure 4E,F).

### 2.5. Plant Hormones

Figure 5 reveals the diverse effects of acetamiprid application during pea seed storage on plant hormone levels throughout pea sprout development. Specifically, IAA levels were reduced, whereas ABA, SA, and JA content exhibited an increase. Moreover, SeNPs and LNTs demonstrated a restorative impact on plant hormone levels.

In comparison to the control group, the IAA content of pea sprouts formed from acetamiprid-treated seeds decreased significantly by 53.5%. However, when SeNPs and LNTs were used in combination, the IAA content increased to a level that was not significantly different from that of the control group (Figure 5A). The SA and ABA contents also increased significantly by 44.7% and 38.5%, respectively. After SeNPs and LNTs were treated individually, both contents decreased significantly but remained significantly higher than those of the control group. However, after treatment with SeNPs and LNTs in combination, the contents were restored to the level of the control group (Figure 5B,C). The JA content exhibited a significant increase of 2.83-fold. Following individual treatment with LNTs, the JA content significantly decreased but remained significantly higher than the control level. Conversely, when combined with SeNPs, the JA content was restored to the control group level. (Figure 5D).

### 2.6. Phenols

As shown in Figure 6A, most phenolic compounds in the edible parts of sprouted pea seeds treated with acetamiprid were significantly reduced compared to the control group.

When used alone or in combination, SeNPs and LNTs could increase the levels of caffeic acid and 4-hydroxybenzoic acid to the levels equivalent to the control group, while there was no significant difference among the three treatments (Figure 6B,E). When SeNPs and LNTs were applied in combination, the contents of kaempferol and luteolin could be restored to the level of the control group. When applied separately, the content of kaempferol was significantly increased but still significantly lower than that of the control group, and the content of luteolin showed no significant difference compared to the acetamiprid treatment group (Figure 6F,H). Additionally, the combined application of SeNPs and LNTs could significantly increase the content of ferulic acid, although the content of ferulic acid was not significantly affected by acetamiprid (Figure 6C). The reparative effect of SeNPs and LNTs on the content of sinapic acid and quercetin was not significant (Figure 6D,G).

### 2.7. Lignin Biosynthesis Pathway

From Figure 7A, it can be observed that the changes in different metabolites in the lignin biosynthesis pathway were not uniform.

After treatment with acetamiprid, the contents of 4-coumaric acid, sinapyl alcohol, 4-coumaryl alcohol, and lignin (Figure 7C,G–I) in germinated pea seeds were significantly increased compared with the control. However, after combined application of SeNPs and LNTs, the contents of sinapyl alcohol, 4-coumaryl alcohol, and lignin could be reduced to a level comparable to the control group (Figure 7G–I), while the content of coumaric acid decreased significantly lower than that of the control group (Figure 7C). Although the treatment of pea seeds with acetamiprid did not cause a significant change in the phenylalanine content of germinated pea sprouts, the application of SeNPs, LNTs, and their combination increased the phenylalanine content by 22.4%, 30.1%, and 35.4%, respectively, compared to the control group (Figure 7B). Compared with the control group, all treatment groups showed a significant decrease in the 4-coumaraldehyde content (Figure 7D), while the content of coniferyl alcohol did not show a significant change (Figure 7F). Compared with the control group, the combined application of SeNPs and LNTs significantly reduced the content of caffeyl alcohol by 22.0%, while there was no significant difference in other treatment groups compared with the control group (Figure 7E).

### 2.8. Correlation Analysis

From the heatmap (Figure 8), it can be seen that there were varying degrees of correlation among the analysis indicators of the edible parts of pea sprouts discussed earlier.

Among them, plant height and edible fresh weight showed a significant positive correlation with IAA, while they exhibited highly significant negative correlations with ABA and SA. Lignin showed a significant or extremely significant positive correlation with SOD, ABA, SA, JA, sinapyl alcohol, and 4-coumaryl alcohol, whereas it exhibited significant or extremely significant negative correlations with plant height and edible fresh weight, amino-acid content, MDA, DPPH^●^-scavenging rate, IAA, caffeic acid, kaempferol, and luteolin. The DPPH^●^-scavenging rate showed significant or extremely significant positive correlations with caffeic acid, kaempferol, and luteolin.

## 3. Discussion

Although pea sprouts are rarely treated with pesticides during production due to their short cultivation cycle, pesticides may be used during pea cultivation [23] and seed storage to control pests [8,9,19]. However, little research has been conducted on the potential impact of pesticide residues on the quality of pea sprouts grown from treated seeds. Acetamiprid is widely recognized as a fast-acting and low-residue insecticide. In this experiment, the half-life of acetamiprid on pea seeds was determined to be 4.18 days (Figure 1), slightly longer than in field [24,25] and greenhouse trials [26,27]. The degradation rate of pesticides is influenced by various factors such as temperature, humidity, light, precipitation, wind, and crop species [28]. The storage environment in this experiment was less complex than that of field and greenhouse conditions because light avoidance treatment and cool and dry storage conditions were applied during the pea seed storage period. Therefore, the half-life of acetamiprid under storage conditions is expected to be longer than that under field and greenhouse conditions. On the 14th day of storage, the residue level of acetamiprid on pea seeds was 0.18 mg/kg, with a degradation rate of 88.6% (Figure 1). After washing, disinfecting, and soaking for 24 h, the residual level of acetamiprid in pea seeds was below 0.01 mg/kg (LOQ), indicating a low dietary intake risk of acetamiprid during pea sprout production.

Despite the fact that more than 88% of acetamiprid residue in pea seeds were degraded during storage, and the residue of acetamiprid following cleaning, disinfecting, and soaking was very low (<0.01 mg/kg), from the perspective of dietary intake risk, the residue of acetamiprid in sprouts during the storage of pea seeds can be considered negligible in terms of human toxicity. However, acetamiprid still had a negative effect on the germination of pea seeds. The plant height and edible fresh weight of pea sprouts treated with acetamiprid were significantly lower than those of the control group (Table 1). Shakir et al. [29] found that the use of insecticides to treat tomato (*Lycopersicon esculentum*) seeds can inhibit germination rates and lead to germination deformities. Braganca et al. [30] also found that residues of pyrethroid pesticides in soil can affect the development of *Cucumis sativus* seedlings. A survey of inhabitants in northwest Mexico [31] showed that sensory attributes of sprouts were an important factor affecting their purchasing decisions. Acetamiprid treatment resulted in poor growth and wilting of pea sprouts, which negatively impacted the sensory experience for consumers (Figure 2B). However, the combined application of SeNPs and LNTs during the production process restored the edible fresh weight to the level of the control group and promoted vigorous growth, characterized by expanded leaves and an improved sensory experience (Figure 2B). Therefore, the use of acetamiprid during pea seed storage can lead to a decrease in yield and sensory quality during the production of pea sprouts. The combined application of SeNPs and mushroom LNTs during pea sprout production can be used as an approach to maintain pea sprout yield and sensory quality.

Germination is a method to improve the nutritional value of seeds, during which macromolecules are hydrolyzed into small molecules, including proteins into amino acids, and polysaccharides into small sugars [32]. Pea seeds treated with acetamiprid showed a significant decrease in amino-acid content during sprout formation (Figure 3C). Zhang et al. [33] found that spraying acetamiprid on cowpea (*Vigna unguiculata* L.) led to a decrease in the content of multiple amino acids in leaves. During the germination of pea seeds, the application of SeNPs and LNTs could, to some extent, restore the content of amino acids; when used together, they could restore the amino-acid content to a level that was not significantly different from that of the control group. Selim et al. [34] demonstrated that the combination of SeNPs and β-aminobutyric acid could increase the amino-acid content in *Medicago interexta* sprouts. Amino acids are nitrogen-containing compounds that can be directly absorbed by the human body. Therefore, the combination of SeNPs and LNTs can be used to prevent the decline in amino-acid content caused by acetamiprid, and it can be considered as a way to maintain the nutritional value of pea sprout production.

As an environmental pollutant, pesticides may cause disturbances in plant reactive oxygen metabolism [11], and many studies [35,36,37] have previously reported the effects of pesticides on plant antioxidant systems. Treatment with acetamiprid resulted in alterations in the antioxidant system of the edible parts of germinated pea seeds. This included significant increases in SOD, CAT, and MDA levels (Figure 4A,C,E), along with notable decreases in POD, GSH, and DPPH^●^-scavenging capacity (Figure 4B,D,F). The catalytic activity of SOD, CAT, POD, and GSH facilitates the conversion of superoxide anions into H_2_O, and MDA formation arises from lipid oxidation, signifying the induction of oxidative stress by acetamiprid in pea sprouts. After treatment with SeNPs and LNTs in combination, all measured antioxidant indicators returned to levels that were not significantly different from the control group. This indicates that SeNPs and LNTs could alleviate the oxidative stress caused by acetamiprid.

Plant hormones are a group of chemical substances produced internally by plants, which can regulate the growth and metabolisms of plants through complex signal transduction networks. Plants respond to environmental stressors, such as pesticide pollution, by regulating various types of plant hormones. Overall, the application of acetamiprid during the storage of pea seeds resulted in plant hormone imbalances in the pea sprouts, characterized by a significant decrease in IAA content (Figure 5A) and a marked increase in ABA, SA, and JA content (Figure 5B–D). IAA is the primary form of endogenous auxin in plants. The biosynthesis and transport of IAA are involved in many growth and development processes, such as embryogenesis, cell division and expansion, elongation of roots and stems, and formation of lateral roots and shoots [38]. After treatment with acetamiprid, the content of IAA was significantly inhibited, and decreases in plant height and edible fresh weight of pea sprouts were observed. When SeNPs and LNTs were coapplied, the content of IAA, plant height, and edible fresh weight were restored to the levels comparable to the control group. From the correlation analysis, the content of IAA showed a significant positive correlation with plant height and edible fresh weight. Therefore, it can be speculated that the residue of acetamiprid may inhibit the growth of pea sprouts by disrupting the biosynthesis of IAA, while SeNPs and LNTs can restore the decrease in plant height and loss of edible fresh weight by restoring the content of IAA. When plants are under stress, plant hormones trigger hormonal crosstalk to respond to the stress. Among these hormones, ABA, SA, and JA play major roles in mediating the defense response of plants to biotic and abiotic stress [39]. Agnihotri et al. [40] pointed that SA may interact with other plant hormones such as JA and ABA to activate the defense mechanisms of *Brassica juncea* L. and alleviate lead toxicity. After treatment with acetamiprid, the contents of JA, SA, and ABA in pea sprouts significantly increased, indicating that pea sprouts may respond to acetamiprid stress by regulating plant hormones. When SeNPs and LNTs were coapplied, the contents of JA, SA, and ABA decreased to levels comparable to the control group, indirectly indicating the alleviating effect of SeNPs and LNTs on acetamiprid stress.

Polyphenols begin to accumulate during the process of seed germination [41,42]. Polyphenols in sprouted legumes are a class of natural bioactive compounds that have been demonstrated to possess various physiological activities and health benefits. They exhibit multiple effects, such as antioxidant, anti-inflammatory, anticancer, antimicrobial, and hypoglycemic activities [43,44]. Additionally, polyphenols in plant edible sprouts can regulate blood lipid and blood pressure levels, improve immunity, promote gastrointestinal health, and prevent cardiovascular diseases [45,46,47]. The disruption of the biosynthesis of plant polyphenols by pesticides has been frequently reported [33,48]. In this study, it was found that the content of multiple polyphenols in the edible parts of germinated pea seeds treated with acetamiprid was significantly reduced, including caffeic acid, sinapic acid, 4-hydroxybenzoic acid, luteolin, quercetin, and rutin. These results suggest that acetamiprid may impair the health-promoting properties of pea sprouts. Fortunately, after the combined application of SeNPs and LNTs, most of the polyphenol indicators could be elevated to the level comparable to the control group, except for sinapic acid. Li et al. [49] found that SeNPs could restore the decrease in polyphenolic compounds in tea leaves caused by imidacloprid. Wang et al. [50] found that the use of LNTs could increase the content of polyphenolic compounds in tobacco seedlings, indicating that SeNPs and LNTs can be used as a means to maintain the content of polyphenolic compounds in crops. Moreover, the polyphenol content in plants can affect their antioxidant capacity [51,52], and the DPPH^●^-scavenging rate of pea sprouts showed a significant or extremely significant positive correlation with caffeic acid, kaempferol, and luteolin, and a nonsignificant positive correlation with other polyphenolic compounds (Figure 7), indicating that the antioxidant capacity of pea sprouts may be related to the content of polyphenolic compounds.

Lignification is the process via which lignin is deposited in the cell walls. The deposition of lignin has a positive effect on plants by improving their mechanical strength, enhancing their resistance to diseases and pests, and hindering the absorption of harmful substances [53]. On the other hand, for fresh-cut vegetables and fruits, undergoing lignification can result in a rough texture, water loss, oxidative browning, and a decrease in palatability and nutritional value, leading to economic losses [54]. Combined with the results of polyphenols, a metabolic pathway map to reveal the effects of SeNPs and LNTs on the lignin synthesis pathway in pea sprouts under acetamiprid toxicity was proposed (Figure 9). Although the bond formation during lignin polymerization is a random chemical process, lignification is a tightly controlled and highly complex biological process [55]. Caffeyl alcohol, coniferyl alcohol, sinapyl alcohol, and 4-coumaryl alcohol are monomers involved in lignin synthesis. Wang et al. [56] indicated that non-melting peach flesh contains higher levels of lignin monomers according to metabolite and transcriptome analyses. Correlation analysis in this study showed a significant positive correlation of lignin content with sinapyl alcohol and 4-coumaryl alcohol in pea sprouts. Lignin content was positively correlated with caffeic acid and coniferyl alcohol although the correlation was not significant. Previous studies [57,58,59] have demonstrated that the accumulation of lignin monomers is accompanied by an increase in the degree of lignification in plants. The signal of reactive oxygen species (ROS) generated by external environmental stress is also one of the reasons for plant lignification [60]. Zuo et al. [61] indicated that nanocomposite packaging delays lignification of *Flammulina velutipes* by regulating ROS metabolisms. Wang et al. [62] demonstrated that ROS-mediated senescence is one of the main reasons for the lignification of king oyster mushrooms (*Pleurotus eryngii*). In this study, the lignin content of pea sprouts showed significant and highly significant correlations with the activity of SOD and the level of GSH, and a significant negative correlation with the level of MDA. The deposition of lignin in plant cell walls is also related to the mediation of plant hormone signaling [63]. In this study, a significant negative correlation was observed between the lignin content of pea sprouts and the content of IAA. On the other hand, a significant or extremely significant positive correlation was observed between the lignin content and the content of ABA, SA, and JA. Cecchetti et al. [64] found that IAA may negatively regulate the biosynthesis of JA, thereby controlling endothecium lignification in *Arabidopsis thaliana*. In this study, a significant negative correlation between IAA and JA could be also observed. Kovacik et al. [65] demonstrated that lignin accumulation in *Matricaria chamomilla* is mediated by SA. Lopez-Orenes et al. [66] showed that the deposition of lignin in the roots of *Zygophyllum fabago* under Pb stress may be related to the priming of SA and the accumulation of ABA. In summary, the lignification of pea sprouts induced by the residue of acetamiprid and the alleviating effects of SeNPs and LNTs on lignification may be related to the levels of lignin monomers, ROS metabolism, and the mediation of plant hormones.

## 4. Materials and Methods

### 4.1. Matericals and Chemicals

Pea seeds were purchased from Jiangsu Suya Food Co., Ltd., (Nanjing, China). A 1500 mg/L selenium nanoparticle solution (mean size: 50–78 nm) was purchased from Jiqi Group Co., Ltd., (Guilin, China). A 2% lentinan aqueous solution was purchased from Aosheng Biological Science and Technology Co., Ltd., (Caoxian, China). Nutrients and antioxidant indicators were analyzed using commercial kits purchased from Comin Biotechnology Co., Ltd., (Suzhou, China). Primary secondary amine (PSA), octadecyl-bonded silica (C_18_), and multiwalled carbon nanotubes (MWCNTs) were purchased from ANPEL Laboratory Technologies Inc., (Shanghai, China). The solvents ethanol, methanol, and formic acid were purchased from Sinopharm Chemical Reagent Co., Ltd., (Shanghai, China). All analytical standards were purchased from Yuanye Bio-Technology Co., Ltd., (Shanghai, China).

### 4.2. Plant Materials and Culture

Pea seeds were placed in a rectangular plastic tray (24.5 cm × 32.5 cm × 9 cm) and evenly spread out. Subsequently, a homogeneous spray of 20 mg/L acetamiprid was applied to the treated group, while the control group received a spray of purified water. The pea seeds were stored in the dark at room temperature for 14 days. The pea seeds were washed with purified water and then disinfected with 0.1% sodium hypochlorite solution for 10 min, followed by two further washes with purified water. After the storage period, the pea seeds were soaked in purified water for 24 h in the dark (water change at 12 h)., and then germinated for 24 h in the dark. Freshly sprouted seeds with consistent growth were selected, placed on petri dishes (30 seeds per petri dish), and placed in an incubator with controlled temperature (25 °C), humidity (70%), and light (continuous dark for 2 days, and 12 h light (3000 lux)/12 h dark cycles for the subsequent 6 days).

Pea seeds sprayed with acetamiprid were sprayed with 5 mg/L selenium nanoparticles (ACESe), 20 mg/L lentinan (ACELNT), or a combination of (ACESeLNT) and purified water (ACE) on the first and third days of transfer to the incubator. Additionally, pea seeds that were not sprayed with acetamiprid were used as the control group (CK) and sprayed with purified water. Each group consisted of six replicates.

After the end of the cultivation period, the growth parameters of pea sprouts were measured. The edible parts of the pea sprouts were collected, ground into powder in liquid nitrogen, and then stored in a refrigerator at −80 °C.

### 4.3. Analysis of Acetamiprid Residues on Pea Seeds

Pea seeds were sampled at 0 (2 h), 3, 5, 7, 10, and 14 days after acetamiprid spraying by adopting the five-point sampling method, with a sampling size of 50 g per time. Additionally, pea seeds were also sampled after soaking for 24 h. The pea seed samples were pulverized into powder by a blender, placed in plastic Ziplock bags, and stored in a −20 °C refrigerator until further analysis.

Samples (2.00 g) were weighed in a 50 mL polypropylene centrifuge tube, to which 10 mL of purified water and 10 mL of acetonitrile were added. The mixtures were vortexed and shaken for 5 min. Then, 3 g of NaCl was added and again vortexed for 5 min. The mixtures were centrifuged at 3800 rpm for 5 min. The supernatant (1 mL) was transferred into a 2 mL centrifuge tube filled with 150 mg of anhydrous magnesium sulfate and 50 mg of octadecylsilane (C_18_), shaken for 2 min, and centrifuged for 2 min at 10,000 rpm. The supernatant was filtered into a vial bottle through a 0.22 μm organic filter membrane. An ultrahigh-performance liquid chromatograph–triple quadrupole mass spectrometer (UPLC–MS/MS, Agilent G6465B, Santa Clara, CA, USA) equipped with a C_18_ column (2.1 mm × 50 mm, 1.8 μm) was utilized to analyze the acetamiprid residue levels. The chromatographic and mass spectrometric parameters are provided in Appendix A, respectively.

According to the standard NY/T 788—2018 [67], the accuracy, precision, and linearity of the method were validated. There was a good linear relationship between the concentration and peak area of acetamiprid in pea seed matrix within the range of 0.01–1.0 mg/L (R^2^ = 0.9971, Appendix A). At the spiking levels of 0.01–2.0 mg/kg, the recoveries were 79–96% with relative standard deviations (RSDs) of 3.7–11.6% (Appendix A). Therefore, this method was stable, reliable, and capable of meeting the requirements for analyzing acetamiprid residue on pea seeds. The lowest spiking level (0.01 mg/kg) was used as the limit of quantification (LOQ) for the instrument.

### 4.4. Nutrients and Antioxidant Indicators

Nutrients including soluble sugar, soluble protein, and amino acid, and antioxidant indicators including peroxidase (POD), superoxide dismutase (SOD), catalase (CAT), glutathione (GSH), malonaldehyde (MDA), and 1,1-diphenyl-2-picrylhydrazyl radical (DPPH^●^) scavenging rate were measured using assay kits. The experimental procedures were rigorously carried out in adherence to the instructions [68] provided by the kits.

### 4.5. Plant Hormone Analysis

The extraction solution was prepared with methanol, purified water, and formic acid mixed at a volume ratio of 80:19:1. Pea sprouts (100 mg) were weighed in a 2 mL centrifuge tube before adding 1 mL of extraction solution. The sample was vortexed for 5 min, sonicated for 30 min, and then centrifuged at 12,000 rpm for 5 min. The extract was transferred to a 2 mL centrifuge tube containing 50 mg of PSA for purifying, vortexed for 5 min, and centrifuged at 12,000 rpm for 5 min. The supernatant was concentrated to dryness by a vacuum concentration meter. The volume was adjusted to 100 μL with the above extraction solution. Lastly, the supernatant was filtered into a vial bottle containing an insert through a 0.22 μm organic filter membrane. The UPLC–MS/MS equipped with a C_18_ column was utilized for analyzing concentrations of phytohormones including indole-3-acetic (IAA), jasmonic acid (JA), salicylic acid (SA), and abscisic acid (ABA). The chromatographic and mass spectrometric parameters are provided in Appendix A, respectively.

### 4.6. Phenolic Compound Analysis

The extraction solution was prepared with ethanol and purified water mixed at a volume ratio of 60:40. Pea sprouts (100 mg) were weighed in a 2 mL centrifuge tube before adding 1 mL of extraction solution. The sample was vortexed for 5 min, sonicated for 30 min, and then centrifuged at 12,000 rpm for 5 min. The extract was transferred to a 2 mL centrifuge tube containing 100 mg of C_18_ for purifying, vortexed for 5 min, and centrifuged at 12,000 rpm for 5 min. The above extraction process was repeated once, and both supernatants were combined. The collected supernatant was concentrated to dryness and adjusted to 1 mL with the above extraction solution. Lastly, the supernatant was filtered into a vial bottle containing an insert through a 0.22 μm organic filter membrane. The UPLC–MS/MS equipped with a C_18_ column was utilized for analyzing the concentrations of phytohormones including caffeic acid, ferulic acid, sinapic acid, 4-hydroxybenzoic acid, kaempferol, quercetin, and luteolin. The chromatographic and mass spectrometric parameters are provided in Appendix A.

### 4.7. Metabolites in Lignin Biosynthesis Analysis

The extraction solution was prepared with methanol, purified water, and formic acid mixed at a volume ratio of 80:19:1. Pea sprouts (100 mg) were weighed in a 2 mL centrifuge tube before adding 1 mL of extraction solution. The sample was vortexed for 5 min, sonicated for 30 min, and then centrifuged at 12,000 rpm for 5 min. The extract was transferred to a 2 mL centrifuge tube containing 5 mg of MWCNTs and 50 mg of C_18_ for purifying, vortexed for 5 min, and centrifuged at 12,000 rpm for 5 min. Finally, the supernatant was filtered into a vial bottle containing an insert through a 0.22 μm organic filter membrane. The UPLC–MS/MS equipped with a Hilic column was utilized for analyzing concentrations of metabolites in lignin biosynthesis including phenylalanine, 4-coumaric acid, 4-coumaraldehyde, caffeyl alcohol, coniferyl alcohol, sinapyl alcohol, and 4-coumaryl alcohol. The chromatographic and mass spectrometric parameters are provided in Appendix A. A high-performance liquid chromatography coupled with an ultraviolet detector (HPLC–UV, Agilent Technologies) equipped with a Hilic column (4.6 mm × 100 mm, 5 μm) was utilized to determine the concentration of lignin. Isocratic elution was carried out for 4.5 min using acetonitrile and purified water (60:40) as the mobile phases, and detection was performed at a wavelength of 217 nm.

### 4.8. Data Process

Statistical analyses were performed with SPSS 23.0 (comparison of means according to Duncan’s test). Graphs were generated using Originlab 2021. Correlation analysis (Pearson’s correlation) was adopted using the Correlation Plots application installed in Originlab 2021.

## 5. Conclusions

This study examined the effects of acetamiprid application during the storage period of pea seeds on subsequent germination. Despite low acetamiprid residue levels on pea seeds after thorough soaking, it still resulted in reduced pea sprout yield, as well as amino-acid and polyphenol content, and it triggered lignin deposition. Co-application of SeNPs and LNTs during germination effectively increased pea sprout yield, as well as amino-acid and polyphenol content, to levels comparable to the control group, while restoring plant hormone levels, antioxidant indicators, and lignin content. These findings provide initial insights into the mechanisms of acetamiprid-induced lignification and the inhibitory effects of SeNPs and LNTs on lignin deposition. The regulation of lignin in pea sprouts appears to be associated with precursor metabolite content, plant hormone mediation, and ROS metabolism. Overall, acetamiprid residue poses a risk to pea sprout yield, nutritional value, and functionality. The combined application of SeNPs and LNTs offers significant mitigation against acetamiprid-induced damage. Future studies are warranted to employ transcriptomics and molecular biology techniques to validate the gene expression levels involved in various metabolic pathways. Such studies will help in making strategies to ensure high yield and quality of pea sprouts. 

## Figures and Tables

**Figure 1 plants-12-02781-f001:**
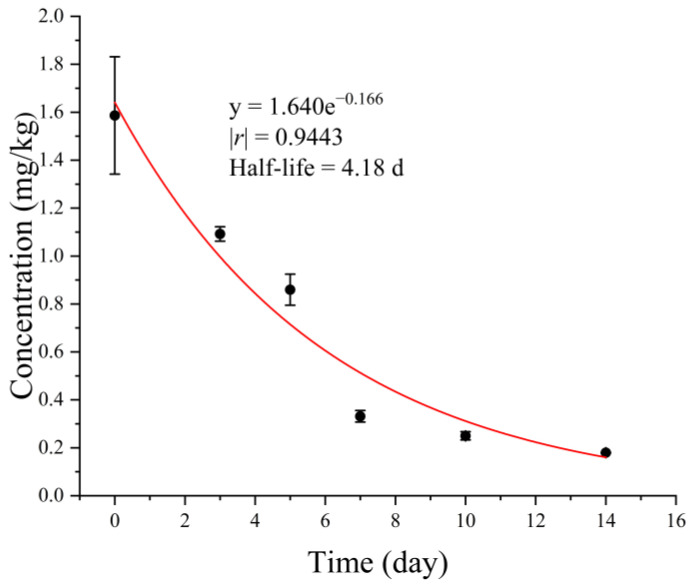
Dissipation dynamic of acetamiprid in pea seeds during the storage period (*n* = 3).

**Figure 2 plants-12-02781-f002:**
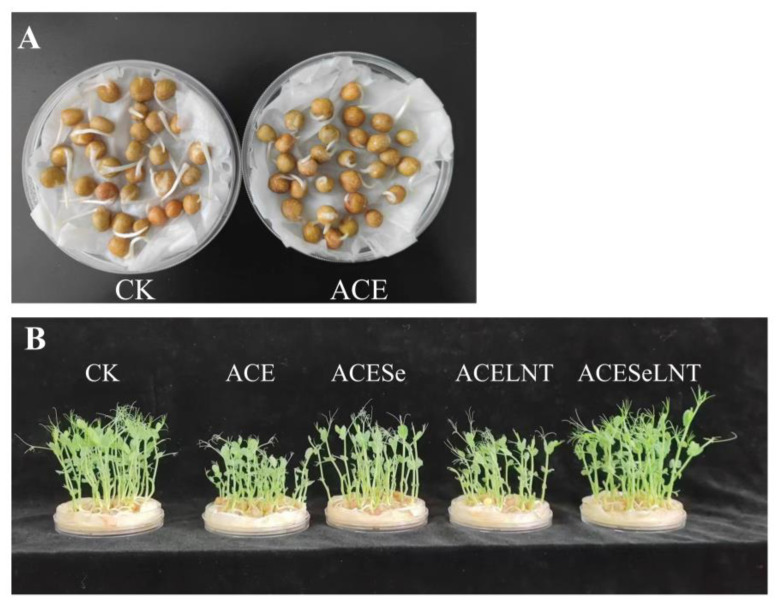
Morphology of germinating pea seeds following acetamiprid treatment and alleviative effects of SeNPs, LNTs, and their combined application. The germination situation on the first day (**A**) and the eighth day (**B**). CK, control; ACE, acetamiprid; ACESe, acetamiprid + Se nanoparticles; ACELNT, acetamiprid + lentinans; ACESeLNT, acetamiprid + Se nanoparticles + lentinans.

**Figure 3 plants-12-02781-f003:**
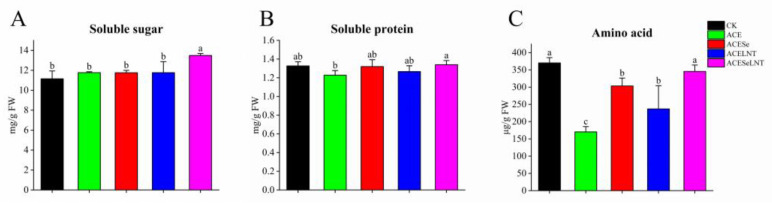
The contents of soluble sugar (**A**), soluble protein (**B**), and amino acid (**C**) in pea sprouts following acetamiprid treatment (ACE) or acetamiprid treated with selenium nanoparticles (ACESe), lentinan (ACELNT), and their combined application (ACESeLNT). Different letters above the columns indicate significant differences (*p* < 0.05) between treatments (*n* = 3). CK, control; ACE, acetamiprid; ACESe, acetamiprid + Se nanoparticles; ACELNT, acetamiprid + lentinans; ACESeLNT, acetamiprid + Se nanoparticles + lentinans.

**Figure 4 plants-12-02781-f004:**
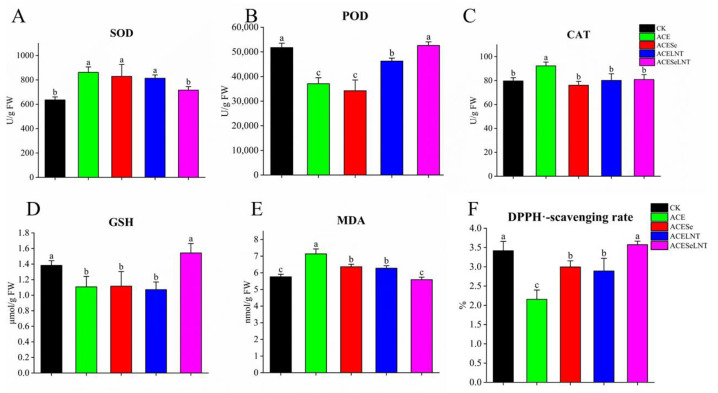
The levels of SOD (**A**), POD (**B**), CAT (**C**), GSH (**D**), MDA (**E**), and DPPH^●^-scavenging rate (**F**) in pea sprouts following acetamiprid treatment (ACE) or acetamiprid treated with selenium nanoparticles (ACESe), lentinan (ACELNT), and their combined application (ACESeLNT). Different letters above the columns indicate significant differences (*p* < 0.05) between treatments (*n* = 3). SOD, superoxide dismutase; POD, peroxidase; CAT catalase, GSH, glutathione; MDA, malondialdehyde; DPPH^●^, 2,2-diphenyl-1-picrylhydrazyl free radical; CK, control; ACE, acetamiprid; ACESe, acetamiprid + Se nanoparticles; ACELNT, acetamiprid + lentinans; ACESeLNT, acetamiprid + Se nanoparticles + lentinans.

**Figure 5 plants-12-02781-f005:**
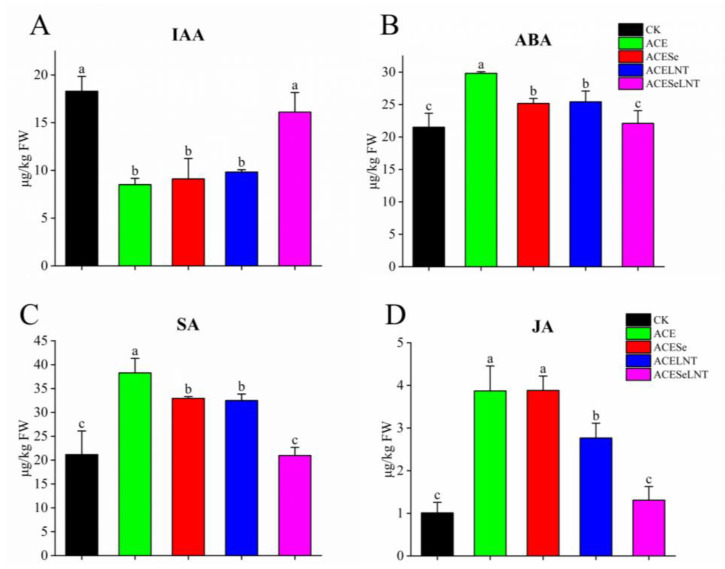
The contents of IAA (**A**), ABA (**B**), SA (**C**), and JA (**D**) in pea sprouts following acetamiprid treatment (ACE)m or acetamiprid treated with selenium nanoparticles (ACESe), lentinan (ACELNT)m and their combined application (ACESeLNT). Different letters above the columns indicate significant differences (*p* < 0.05) between treatments (*n* = 3). IAA, indole-3-acetic acid; JA, jasmonic acid; SA, salicylic acid; CK, control; ACE, acetamiprid; ACESe, acetamiprid + Se nanoparticles; ACELNT, acetamiprid + lentinans; ACESeLNT, acetamiprid + Se nanoparticles + lentinans.

**Figure 6 plants-12-02781-f006:**
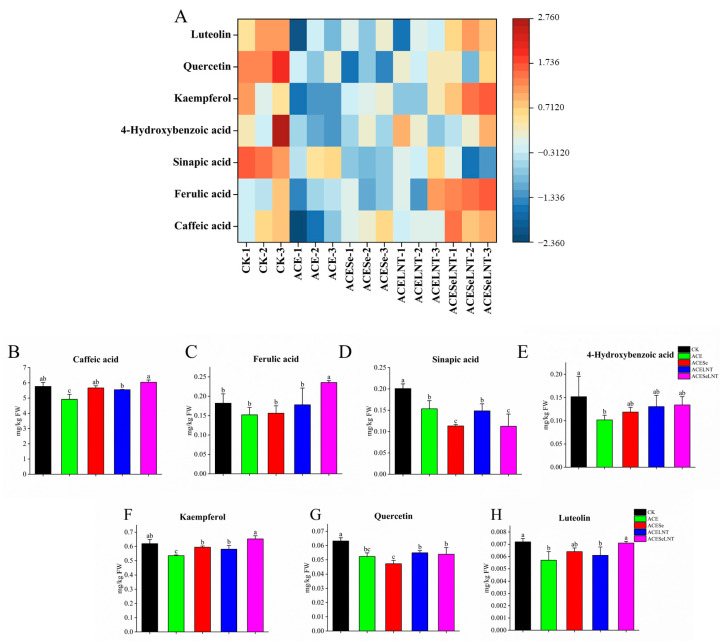
The contents of caffeic acid (**B**), ferulic acid (**C**), sinapic acid (**D**), 4-hydroxybenzoic acid (**E**), kaempferol (**F**), quercetin (**G**), and luteolin (**H**) in pea sprouts following acetamiprid treatment (ACE) or acetamiprid treated with selenium nanoparticles (ACESe), lentinan (ACELNT), and their combined application (ACESeLNT). (**A**) Heatmap for visualizing the alteration of phenolic compounds. Different letters above the columns indicate significant differences (*p* < 0.05) between treatments (*n* = 3). CK, control; ACE, acetamiprid; ACESe, acetamiprid + Se nanoparticles; ACELNT, acetamiprid + lentinans; ACESeLNT, acetamiprid + Se nanoparticles + lentinans.

**Figure 7 plants-12-02781-f007:**
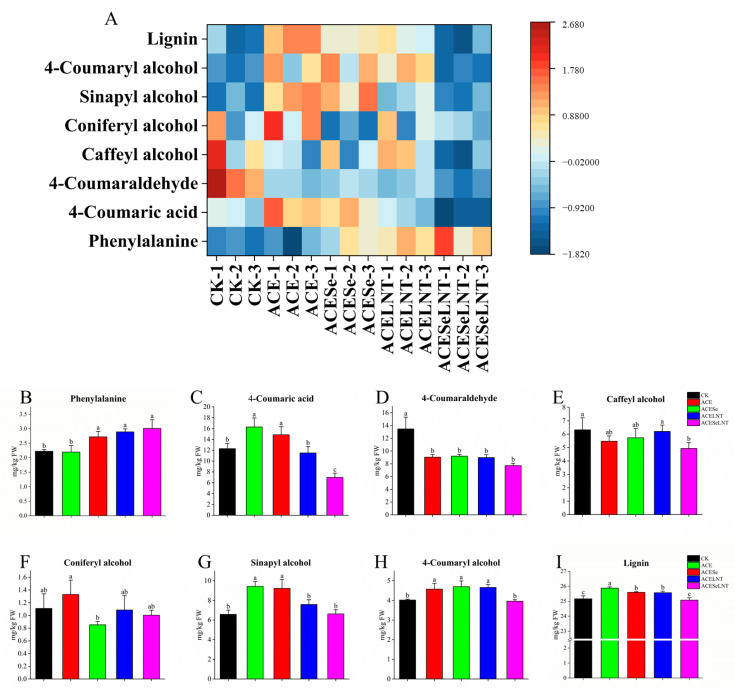
The contents of phenylalanine (**B**), 4-coumaric acid (**C**), 4-coumaraldehyde (**D**), caffeyl alcohol (**E**), coniferyl alcohol (**F**), sinapyl alcohol (**G**), 4-coumaryl alcohol (**H**), and lignin (**I**) in pea sprouts following acetamiprid treatment (ACE) or ACE treated with selenium nanoparticles (ACESe), lentinan (ACELNT) and their combined application (ACESeLNT). (**A**) Heatmap for visualizing the alteration of metabolites in the lignin biosynthesis pathway. Different letters above the columns indicate significant differences (*p* < 0.05) between treatments (*n* = 3). CK, control; ACE, acetamiprid; ACESe, acetamiprid + Se nanoparticles; ACELNT, acetamiprid + lentinans; ACESeLNT, acetamiprid + Se nanoparticles + lentinans.

**Figure 8 plants-12-02781-f008:**
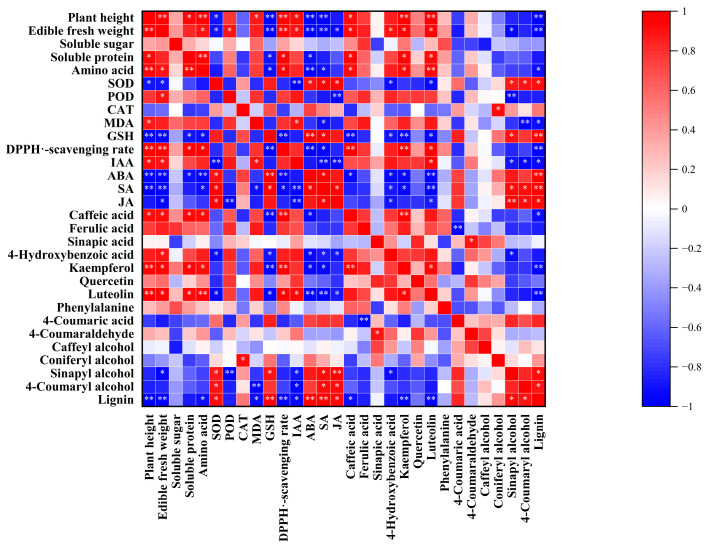
A heatmap was generated to illustrate the correlations and significance among the analysis parameters of edible parts in pea sprouts. Color intensity indicates correlation strength, and color signifies correlation direction. * Significant correlation (*p* < 0.05); ** extremely significant correlation (*p* < 0.01). SOD, superoxide dismutase; POD, peroxidase; CAT, catalase; MDA, malondialdehyde; GSH, glutathione; DPPH^●^, 2,2-diphenyl-1-picrylhydrazyl free radical; IAA, indole-3-acetic acid; ABA, abscisic acid; SA, salicylic acid; JA, jasmonic acid.

**Figure 9 plants-12-02781-f009:**
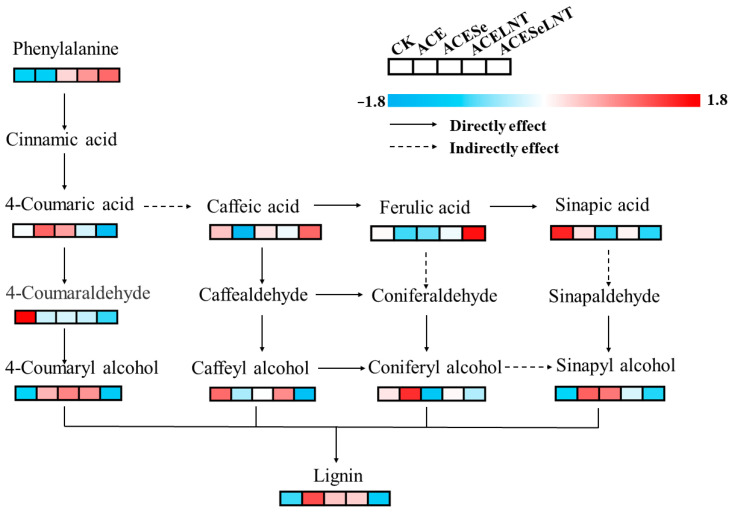
Effects of SeNPs and LNTs on the lignin biosynthesis pathway in pea sprouts under acetamiprid toxicity.

**Table 1 plants-12-02781-t001:** Biomass of pea sprouts following SeNPs, LNTs, and their combined applications.

Treatment	Plant Height (cm) *n* = 12	Root Length (cm) *n* = 12	Edible Fresh Weight (g) *n* = 6	Root Fresh Weight (g) *n* = 6
CK	7.67 ± 0.28 a	5.89 ± 1.60 a	4.39 ± 0.57 a	0.78 ± 0.16 a
ACE	5.32 ± 0.38 d	5.24 ± 1.13 a	2.86 ± 0.22 c	0.70 ± 0.16 a
ACESe	6.53 ± 0.39 b	5.90 ± 1.10 a	3.47 ± 0.25 b	0.82 ± 0.14 a
ACELNT	6.12 ± 0.24 c	5.22 ± 1.58 a	3.73 ± 0.19 b	0.79 ± 0.09 a
ACESeLNT	7.76 ± 0.32 a	5.66 ± 1.21 a	4.59 ± 0.20 a	0.72 ± 0.16 a

Note: The means ± SD displayed in each column followed by different letters are significantly different at *p* < 0.05.

## Data Availability

The data presented in this study are available on request from the corresponding author.

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
