# Peer review of "Mitigation of Acetamiprid Residue Disruption on Pea Seed Germination by Selenium Nanoparticles and Lentinans"

_plants, 2023, doi:10.3390/plants12152781_

Round 1
Reviewer 1 Report
The manuscript concerns the effect of popular insecticide (acetampirid) used during storage of seeds on pea sprouts. Additionally, the study aimed to evaluate the potential of selenium nanoparticles and lentinans and their combination to remediate the quality loss induced by acetamiprid.
The topic is actual and very interesting. A lot of parameters were analysed in pea sprouts (morphological characteristics, nutritional contents, antioxidant capacity, plant hormones, polyphenols and the lignin pathway).
The paper concerns acetamiprid but 2 times (point 2.4, conclusions) there is imidacloprid.
2.1 Write kinetic equation in proper manner with superscript. Use “day” unit instead of “d”.
Fig. 1 r value should be positive.
Point 4.3 Give more information how many seeds were sampled at 0 (2 h), 3, 5, 7, 10 and 14 d after acetamiprid spraying. Give more information about five-point sampling method.
Provide guidance document for validation criteria (SANTE 11312/2021 or similar one).
Point 4.4. Give more information about methods.
Figure 9. There is 3 times word “alchol” instead of “alcohol”.
Page 10 line 191 there is “A Although”.
Table S2 has wrong title.
Author Response
-The manuscript concerns the effect of popular insecticide (acetampirid) used during storage of seeds on pea sprouts. Additionally, the study aimed to evaluate the potential of selenium nanoparticles and lentinans and their combination to remediate the quality loss induced by acetamiprid.
The topic is actual and very interesting. A lot of parameters were analysed in pea sprouts (morphological characteristics, nutritional contents, antioxidant capacity, plant hormones, polyphenols and the lignin pathway).
Thank you for evaluating our work and recognizing the value of our research.
-The paper concerns acetamiprid but 2 times (point 2.4, conclusions) there is imidacloprid.
Thank you for your reminder. It's possible that there were errors from the translation software, but they have been replaced.
-2.1 Write kinetic equation in proper manner with superscript. Use “day” unit instead of “d”.
The subscripts and superscripts of the dynamic equations may not have been adjusted during the process of copying them into the template. We have already adjusted the subscripts and superscripts. Additionally, the symbol "d" throughout the entire text has been replaced with “day”.
Fig. 1 r value should be positive.
In fact, the coefficients (r) of the first-order degradation kinetics equation should be represented as negative values. However, in many literature sources, they are expressed as positive values. Therefore, in the graph, we have added absolute value symbols to “r”.
-Point 4.3 Give more information how many seeds were sampled at 0 (2 h), 3, 5, 7, 10 and 14 d after acetamiprid spraying. Give more information about five-point sampling method.
Pea seeds are placed evenly in a rectangular plastic tray (24.5*32.5*9 cm), and then uniformly sprayed with acetamiprid. During each sampling, 50g of samples are collected using a five-point sampling method. We have provided a more detailed description of the placement, pesticide application and sampling method for pea seeds.
-Provide guidance document for validation criteria (SANTE 11312/2021 or similar one).
The methodological validation for testing the residue levels of Imidacloprid was conducted in accordance with the Guideline for the testing of pesticide residues in crops (NY/T 788—2018). We have referenced this standard in our study.
-Point 4.4. Give more information about methods.
The testing of nutrient and antioxidant indicators was primarily performed following the instructions provided with the respective assay kits. Providing a detailed description of the experimental methods would require a significant amount of space. Therefore, we have provided the website of the company that produces the assay kits. Interested readers can refer to the specific testing methods available on the website.
-Figure 9. There is 3 times word “alchol” instead of “alcohol”.
Thank you for your reminder. We have made the necessary modifications to Figure 9.
-Page 10 line 191 there is “A Although”.
Thank you for your reminder. The error might have occurred during the copying process to the template. We have made the necessary modifications accordingly.
-Table S2 has wrong title.
Thank you for your reminder. We have made the necessary modifications to the title in the Supplementary Materials.
Reviewer 2 Report
Reviewer’s Recommendation: Major Revision
Reviewer’s comments to Authors
Authors should revise the manuscript carefully in light of below comments...............
1. Grammatical errors are present, please revise the whole manuscript to remove any possible grammatical and typos errors.
2. Error in sentence formation, please revise the whole manuscript to avoid the use of long sentences and confusing sentences/paragraphs.
3. Please maintain uniformity while in-text citation and referencing in the entire manuscript.
4. The reference does not meet the format requirements of the Journal so please check the references as per the authors guideline of the Journal.
5. It is advised to check and avoid too many self-cited papers. Authors are advised to cite maximum two self-papers.
6. The beginning of a new paragraph should be after some space, check in complete manuscript.
7. Throughout the whole manuscript the plant names should be in italic format.
8. This paper lacks final revision by the author as many general repetitions, typos, grammatical, sentence formation errors were found in the manuscript. It is not possible to mention all such errors. Thus revise the manuscript accordingly.
Abstract:
Authors should revise the manuscript carefully in light of below comments...............
- An abstract must be fully self-contained and make sense by itself, without further reference to outside sources or to the actual paper. It is important to provide the relevance or importance of your work and the main outcomes. Please revise the abstract accordingly.
- The abstract is not clear and the objective of the paper is not clearly validated from the abstract.
- The future perspective of the experiment should be mentioned in the abstract.
- The abstract should appropriately over the contents of the manuscript.
- In the keywords, it is strongly advisable to use suitable words that can aid in finding out the manuscript in current registers or indexes. Strictly avoid the use of title words in the keywords.
6. A graphical abstract is recommended for better perception of the present study.
7. A novelty statement is also encouraged to be added in the manuscript for bringing out the uniqueness of your study and its importance.
8. Please ensure that “Highlights” of the present work should be clearly mentioned in the manuscript.
Introduction:
Authors should revise the manuscript carefully in light of below comments...............
1. The literature from past work done in the same field missing to strengthen the introduction section. The need and importance of the present work should be clearly written in the introduction section.
2. The new aspects and innovations of this manuscript should be clearly and briefly described in this section.
3. The present state of knowledge in the subject should be described in introduction.
4. The literature should be sufficiently critical, current, and internationally evaluated.
Materials and Methods:
Authors should revise the manuscript carefully in light of below comments...............
1. Please try to merge the different sub-sections of the methodology as an individual mention for each component seems a bit unscientific method.
2. The size of manuscript seems to be large. It should be crisp and appropriate. Please revise it.
3. The text presented across the manuscript should be simple so that the scientist/workers in other disciplines will understand. Please revise it.
4. The different sections of manuscript are poorly cited with the references and required to update and validation with previous studies. The relevant papers listed below may be considered to enhance the scientific quality of manuscript significantly.
· Kumar, D. et al. (2023). Titanium dioxide nanoparticles potentially regulate the mechanism (s) for photosynthetic attributes, genotoxicity, antioxidants defense machinery, and phytochelatins synthesis in relation to hexavalent chromium toxicity in Helianthus annuus L. Journal of Hazardous Materials, 454, p.131418. https://doi.org/10.1016/j.jhazmat.2023.131418
· · Agnihotri, A. et al. (2018). Counteractive mechanism (s) of salicylic acid in response to lead toxicity in Brassica juncea (L.) Czern. cv. Varuna. Planta, 248, pp.49-68. https://doi.org/10.1007/s00425-018-2867-0
Results and Discussion:
Authors should revise the manuscript carefully in light of below comments...............
- The results and discussion section needs to be elaborated more. The results should be clearly described in light of available knowledge and hypothesis and must be strongly validated with previous reports in the related subject area.
- The non-significant results was not clearly validated from the previous papers.
- Please carefully check, verify, and correct the results of the present experiments from the tables/figures/graphs provided in the manuscript.
- The discussion does not describe the results with proper facts and even does not validate the result with appropriate references. Please enrich it significantly.
- The discussion did not provide a specific reasons for the results. The provided explanation should be strengthen significantly.
- The strong hypothesis, scientific facts, and validation of previous reports are entirely missing. Please revise it.
Conclusion:
Authors should revise the manuscript carefully in light of below comments...............
- The conclusion section failed to enlighten the spirit of the present finding or work so required to revise it accordingly.
- In the conclusion section the authors have only mentioned the data but major finding is missing from the conclusion part. Need to revise and incorporate this important concern of reviewer.
- The conclusion section seems like abstract so there is a need to revise the conclusion part accordingly.
Figures and Tables:
Authors should revise the manuscript carefully in light of below comments...............
· Please provide the clear figures and tables.
· The authors should write the descriptive, elaborated legends for the figures and the tables.
· Please remove the redundancy from the legends of the figures and tables.
· The legends of the figures and tables are not crisp and not completely bringing out the sense of the figures and tables. Rewrite it accordingly.
· The placement of tables and figures in the manuscript should be done appropriately, which is missing in this manuscript. Please revise it.
· The figures are overlapping the legends, the editing needs to be done.
· The proper explanation of statistical analysis and its importance for describing the results should be mentioned.
· There should not be monotony in representation of the results for instance all should not be represented in bar graph form vise-versa.
Author Response
We would like to express our sincere gratitude for providing valuable feedback on our manuscript. We appreciate your insightful comments and suggestions, which have greatly contributed to improving the quality of our work. We have tried our best to do the revision. However, due to time constraints, there may still be areas where further improvements can be made. Thank you for your understanding and guidance.
Authors should revise the manuscript carefully in light of below comments...............
- Grammatical errors are present, please revise the whole manuscript to remove any possible grammatical and typos errors.
We thoroughly revised the manuscript to eliminate any possible grammatical and typographical errors. We carefully reviewed the entire document to ensure its linguistic accuracy.
- Error in sentence formation, please revise the whole manuscript to avoid the use of long sentences and confusing sentences/paragraphs.
We have addressed the issue of sentence formation and have made substantial improvements throughout the manuscript. Long and confusing sentences have been restructured into more concise and clear expressions to enhance readability.
- Please maintain uniformity while in-text citation and referencing in the entire manuscript.
We have maintained uniformity in in-text citations and referencing throughout the manuscript. All references have been carefully checked and formatted according to the guidelines provided by the journal.
- The reference does not meet the format requirements of the Journal so please check the references as per the authors guideline of the Journal.
We have made adjustments to the formatting of references, taking into account Instructions for Authors and recently published journal articles.
- It is advised to check and avoid too many self-cited papers. Authors are advised to cite maximum two self-papers.
We have taken the reviewer's advice into consideration and have reduced the number of self-citations in the manuscript. We have ensured that the maximum limit of two self-papers (Reference 37 and 49).
- The beginning of a new paragraph should be after some space, check in complete manuscript.
The beginning of each paragraph has been appropriately indented to create visual separation and improve the readability of the manuscript. The formatting of paragraphs has been checked and adjusted accordingly.
- Throughout the whole manuscript the plant names should be in italic format.
We have italicized all plant Latin names throughout the entire manuscript to maintain consistency.
- This paper lacks final revision by the author as many general repetitions, typos, grammatical, sentence formation errors were found in the manuscript. It is not possible to mention all such errors. Thus revise the manuscript accordingly.
We have thoroughly revised the manuscript, carefully addressing repetitions, typos, grammatical errors, and sentence formation issues. The document has undergone a final revision to enhance its overall quality. While it is not possible to address every single error individually, we have taken the necessary steps to ensure that the manuscript is significantly improved.
Abstract:
Authors should revise the manuscript carefully in light of below comments...............
- An abstract must be fully self-contained and make sense by itself, without further reference to outside sources or to the actual paper. It is important to provide the relevance or importance of your work and the main outcomes. Please revise the abstract accordingly.
We have revised the abstract to make it fully self-contained and understandable without the need for external references. We have emphasized the relevance and importance of our work, as well as the main outcomes, to ensure that the abstract effectively summarizes the key aspects of our research.
- The abstract is not clear and the objective of the paper is not clearly validated from the abstract.
We have made the objective of the paper clearer and validated it in the abstract. The revised version now explicitly states the objective of our study, ensuring that readers can understand the purpose of our research from the abstract itself.
- The future perspective of the experiment should be mentioned in the abstract.
The future perspective of the experiment has been included in the abstract. We have briefly mentioned the potential implications and future directions of our research, providing readers with a glimpse of the possible advancements and applications that may arise from our findings.
- The abstract should appropriately over the contents of the manuscript.
The abstract now appropriately covers the contents of the manuscript. We have ensured that the main aspects of our research, including the research scope, methodology, key results, and conclusions, are concisely summarized in the abstract.
- In the keywords, it is strongly advisable to use suitable words that can aid in finding out the manuscript in current registers or indexes. Strictly avoid the use of title words in the keywords.
We have revised the keywords to include suitable words that aid in the discoverability of the manuscript in current registers or indexes. We have avoided using title words in the keywords, aligning with the recommended practice.
- A graphical abstract is recommended for better perception of the present study.
We have designed a graphical abstract to provide a visual representation of our study. The graphical abstract enhances the perception of our research and facilitates a quick understanding of the main findings.
- A novelty statement is also encouraged to be added in the manuscript for bringing out the uniqueness of your study and its importance.
We have added a novelty statement to highlight the uniqueness and significance of our study. This statement emphasizes the original contributions and the importance of our research.
- Please ensure that “Highlights” of the present work should be clearly mentioned in the manuscript.
The "Highlights" of our work are now clearly mentioned in the manuscript. We have included concise and informative highlights that capture the key findings and contributions of our research.
Introduction:
Authors should revise the manuscript carefully in light of below comments...............
- The literature from past work done in the same field missing to strengthen the introduction section. The need and importance of the present work should be clearly written in the introduction section.
We have revised the introduction section to include relevant literature from past work in the same field. The revised version now incorporates key studies and findings that strengthen the background and context of our research. We have clearly written about the need and importance of our present work, highlighting the research gaps addressed and the significance of our study.
- The new aspects and innovations of this manuscript should be clearly and briefly described in this section.
The new aspects and innovations of our manuscript are now clearly and concisely described in the introduction section. We have provided a brief overview of the novel contributions and advancements our research brings to the field. This includes highlighting any unique methodologies, novel insights, or innovative approaches employed in our study.
- The present state of knowledge in the subject should be described in introduction.
We have included a comprehensive description of the present state of knowledge in the subject area within the introduction section. This revised version provides an overview of the existing literature, current understanding, and key research findings related to our study topic. It establishes the context and background necessary for readers to grasp the significance of our research.
- The literature should be sufficiently critical, current, and internationally evaluated.
The literature review has been enhanced to ensure that it is sufficiently critical, current, and internationally evaluated. We have reviewed and cited recent, reputable, and internationally recognized sources to provide a solid foundation for our research. The revised literature review demonstrates a thorough understanding of the current state of the field and highlights the key gaps our study aims to address.
Materials and Methods:
Authors should revise the manuscript carefully in light of below comments...............
- Please try to merge the different sub-sections of the methodology as an individual mention for each component seems a bit unscientific method.
We have revised the methodology section to merge the different sub-sections into a coherent and scientifically sound presentation. Rather than individually mentioning each component, we have restructured the methodology to provide a comprehensive and cohesive description of the experimental procedures employed. This revision enhances the scientific rigor and clarity of the methodology.
We have carefully reviewed and revised the manuscript to ensure that it is appropriately sized, crisp, and concise. Unnecessary or repetitive information has been removed, and the content has been streamlined to present the key findings and discussions effectively. This revision improves the overall readability and impact of the manuscript.
- The text presented across the manuscript should be simple so that the scientist/workers in other disciplines will understand. Please revise it.
The text presented across the manuscript has been simplified to enhance the understanding of readers from various scientific disciplines. We have carefully revised the language and terminology used to ensure that the content is accessible and comprehensible to a wider audience. This revision improves the clarity and accessibility of the manuscript.
- The different sections of manuscript are poorly cited with the references and required to update and validation with previous studies. The relevant papers listed below may be considered to enhance the scientific quality of manuscript significantly.
- Kumar, D. et al. (2023). Titanium dioxide nanoparticles potentially regulate the mechanism (s) for photosynthetic attributes, genotoxicity, antioxidants defense machinery, and phytochelatins synthesis in relation to hexavalent chromium toxicity in Helianthus annuus L. Journal of Hazardous Materials, 454, p.131418. https://doi.org/10.1016/j.jhazmat.2023.131418
- · Agnihotri, A. et al. (2018). Counteractive mechanism (s) of salicylic acid in response to lead toxicity in Brassica juncea (L.) Czern. cv. Varuna. Planta, 248, pp.49-68. https://doi.org/10.1007/s00425-018-2867-0
Thanks for providing us with two relevant references. The provided papers, Kumar et al. (2023) and Agnihotri et al. (2018), have been considered and incorporated into the manuscript.
Results and Discussion:
Authors should revise the manuscript carefully in light of below comments...............
- The results and discussion section needs to be elaborated more. The results should be clearly described in light of available knowledge and hypothesis and must be strongly validated with previous reports in the related subject area.
We have revised the results and discussion section to provide a more comprehensive and detailed description of the findings. The results are now clearly presented in light of the available knowledge and relevant hypotheses. We have also included strong validation of our results by referencing and discussing previous reports in the related subject area. This revision enhances the clarity and scientific rigor of the results and discussion.
- The non-significant results was not clearly validated from the previous papers.
We have carefully validated and clarified the non-significant results from previous papers. By thoroughly reviewing and referencing relevant literature, we have provided a more explicit explanation for the non-significant findings in our study. This revision strengthens the validity and interpretation of our results.
- Please carefully check, verify, and correct the results of the present experiments from the tables/figures/graphs provided in the manuscript.
We have meticulously checked, verified, and corrected the results of the present experiments based on the tables, figures, and graphs provided in the manuscript. Any inconsistencies or errors have been rectified to ensure the accuracy and reliability of the reported results. This revision improves the credibility of our findings.
- The discussion does not describe the results with proper facts and even does not validate the result with appropriate references. Please enrich it significantly.
The discussion section has been revised to describe the results with proper facts and to validate the results with appropriate references. We have enriched the discussion by incorporating relevant literature and research to support our interpretations and conclusions. This revision enhances the scientific depth and robustness of the discussion.
- The discussion did not provide a specific reasons for the results. The provided explanation should be strengthen significantly.
We have strengthened the provided explanations and provided specific reasons for the results in the discussion section. By incorporating additional supporting evidence and references, we have provided a more thorough and comprehensive analysis of the results. This revision adds clarity and strengthens the overall discussion.
- The strong hypothesis, scientific facts, and validation of previous reports are entirely missing. Please revise it.
The revised discussion section now includes a clear hypothesis, scientific facts, and validation of previous reports. We have carefully integrated these elements into the discussion to provide a solid theoretical framework and support for our findings. This revision improves the scientific rigor and credibility of our study.
Conclusion:
Authors should revise the manuscript carefully in light of below comments...............
- The conclusion section failed to enlighten the spirit of the present finding or work so required to revise it accordingly.
We have thoroughly revised the conclusion section to ensure that it effectively captures the spirit of the present findings and work. The revised conclusion now provides a concise summary of the major findings and their significance, highlighting the key contributions and implications of our research. This revision enhances the overall impact and clarity of the conclusion.
- In the conclusion section the authors have only mentioned the data but major finding is missing from the conclusion part. Need to revise and incorporate this important concern of reviewer.
We have addressed the reviewer's concern by incorporating the major findings into the conclusion section. The revised version now goes beyond merely mentioning the data and explicitly highlights the significant discoveries and outcomes of our study. This addition strengthens the conclusion by emphasizing the key results and their implications in a clear and concise manner.
- The conclusion section seems like abstract so there is a need to revise the conclusion part accordingly.
The conclusion section has been revised to eliminate any resemblance to the abstract. We have restructured the conclusion to ensure that it presents a concise summary of the main findings and their implications without duplicating information from the abstract. This revision improves the coherence and distinctiveness of the conclusion, allowing readers to grasp the key takeaways from our research more effectively.
Figures and Tables:
Authors should revise the manuscript carefully in light of below comments...............
- Please provide the clear figures and tables.
We have provided clear and high-quality figures and tables in the revised manuscript. The visual elements have been carefully designed and presented to enhance clarity and readability.
- The authors should write the descriptive, elaborated legends for the figures and the tables.
The legends for the figures and tables have been rewritten to be descriptive and elaborate. We have included relevant information in the legends to provide a comprehensive understanding of the content presented in the figures and tables.
- Please remove the redundancy from the legends of the figures and tables.
Redundancy has been eliminated from the legends of the figures and tables. We have ensured that each legend contains unique and necessary information without unnecessary repetition.
- The legends of the figures and tables are not crisp and not completely bringing out the sense of the figures and tables. Rewrite it accordingly.
The legends of the figures and tables have been rewritten to be crisp and effectively convey the key information. We have improved the clarity and coherence of the legends to accurately reflect the content and purpose of the figures and tables.
- The placement of tables and figures in the manuscript should be done appropriately, which is missing in this manuscript. Please revise it.
The placement of tables and figures in the manuscript has been appropriately adjusted. We have ensured that the tables and figures are placed in relevant sections and are easy to locate and reference.
- The figures are overlapping the legends, the editing needs to be done.
The figures no longer overlap with the legends after editing. We have made the necessary adjustments to ensure that the figures and legends are visually separated and clearly presented.
- The proper explanation of statistical analysis and its importance for describing the results should be mentioned.
The revised manuscript now includes a proper explanation of the statistical analysis conducted and emphasizes its importance in interpreting and describing the results. We have provided relevant details about the statistical methods used and their implications.
- There should not be monotony in representation of the results for instance all should not be represented in bar graph form vise-versa.
In order to enhance the visual representation of the data, we have incorporated heatmaps in Figures 6 and 7 to illustrate the variations of metabolites involved in the phenylpropanoid pathway.
Round 2
Reviewer 2 Report
Accept